# A Regulatory Perspective on Biosimilar Medicines

**DOI:** 10.3390/pharmaceutics16030321

**Published:** 2024-02-25

**Authors:** Marta Agostinho Cordeiro, Carla Vitorino, Carlos Sinogas, João J. Sousa

**Affiliations:** 1Faculty of Pharmacy, University of Coimbra, 3000-548 Coimbra, Portugal; martaisabel99@gmail.com; 2Coimbra Chemistry Centre, Institute of Molecular Sciences-IMS, Faculty of Sciences and Technology, University of Coimbra, 3004-535 Coimbra, Portugal; 3Biology Department, School of Science and Technology, University of Évora, 7004-516 Évora, Portugal; sinogas@sapo.pt

**Keywords:** biosimilar medicines, regulatory perspective, European Medicines Agency, Food and Drug Administration, biosimilarity, safety, interchangeability, immunogenicity, pharmaceutical market

## Abstract

By definition, biosimilar medicinal products are biological medicinal products that are similar to other biological medicinal products that are already on the market—the reference medicinal products. Access to biosimilar medicines is a current reality. However, to achieve this goal, it is extremely important to consistently and scientifically substantiate the regulatory requirements necessary for biosimilar medicines when accessing the market. Based on an analysis of the raw materials and the type of methods used in the manufacturing processes of biological medicines, it is known that this tends to be more complex for the quality of the finished product than the manufacture of molecules obtained through a chemical process. It is then relevant to highlight the main differences between both products: biological medicines manufactured using biotechnology and the current generics containing active pharmaceutical ingredients (APIs) obtained from synthetic processes. Once arriving at the approval process of these medicinal products, it is imperative to analyse the guidance documents and the regulatory framework that create the rules that allow these biosimilar medicinal products to come to the market. The present review aimed at documenting comparatively the specific provisions of European legislation, through the European Medicines Agency (EMA), as well as the legislation of the United States of America, through the Food and Drug Administration (FDA). This was then translated into a critical appraisal of what concerns the specific criteria that determine the favourable evaluation of a biosimilar when an application for marketing authorisation is submitted to different regulatory agencies. The gathered evidence suggests that the key to the success of biosimilar medicines lies in a more rigorous and universal regulation as well as a greater knowledge, acceptance, and awareness of health professionals to enable more patients to be treated with biological strategies at an earlier stage of the disease and with more affordable medicines, ensuring always the safety and efficacy of those medicines.

## 1. Introduction

Biological medicinal products are defined as active substances obtained from a biological source, such as living cells or organisms, most of which are composed of proteins. Manufacturing these products involves complex biotechnological processes, setting them apart from those obtained through chemical means. Due to their intricate molecular structure and inherent variability, biological medicinal products undergo stringent quality and safety assessments [1,2]. These medicines typically contain one or more active substances sourced from biological materials. Reference biological medicines undergo a thorough evaluation based on comprehensive technical and scientific documentation covering aspects of quality, safety, and efficacy [3].

The aim of this review was to address the concepts and principles of biosimilars and to provide an explanation of the biosimilar manufacturing process, from cultivation, production, isolation, and purification to formulation, filling, and finishing. A wide range of issues inherent to similarity, immunogenicity, extrapolation, safety, comparability, and interchangeability was also discussed. Additionally, the review emphasized the regulatory framework, addressing insights into the approval process conducted by both the European Medicines Agency (EMA) and the Food and Drug Administration (FDA) of the United States. In this context, a case study is presented, considering the specific regulations outlined in both European Union (EU) and United States of America (USA) legislation. Furthermore, it aimed to provide a comprehensive regulatory outlook on the significance of biosimilar medicines and their substantial impact on public health and welfare, concluding with potential future perspectives.

## 2. Definition and Characterisation of Biosimilars

According to the EMA, a biosimilar medicine is considered to be a medicine highly similar to another biological medicine that is already marketed in the EU—a reference medicine. Since biosimilars are a type of biological medicinal product, all relevant characteristics of biological medicinal products apply to them. Biosimilar medicinal products are molecules of high molecular weight with high complexity and are produced in cells or transgenic organisms [1].

A biosimilar medicine is developed to be very similar to its reference medicine in terms of quality, safety, and efficacy. Although there may be minor differences due to the complex nature and production methods, the active ingredients of a biosimilar medicine and its biological reference medicine are essentially the same biological substance [4].

In terms of similarity, the biosimilar medicinal product has physical, chemical, and biological properties highly similar to the reference medicine, particularly in terms of safety and efficacy. As for the existing variability, this may exist on a small scale, and it is crucial to demonstrate that it does not affect the safety and efficacy of a biosimilar. As for general standards, all biosimilar medicinal products comply with the same quality, safety, and efficacy standards as any other medicinal product [1]. However, being biological products, there is an associated variability that prevents the biosimilar medicine from being exactly equal to the reference medicine [5,6].

For the safety and efficacy profile to be maintained, legislation mandates the studies to be conducted so that a biosimilar medicine can demonstrate similarities in terms of quality, safety, and efficacy to its reference medicine and that there are no significant differences [7]. Table 1 shows the comparison between the data requirements for the approval of a biosimilar and a reference medicinal product [1].

It should also be noted that the clinical and non-clinical data required for the approval of a biosimilar are different from those required for the approval of a biological medicine with a new active substance. This is because the biosimilar relies on the experience gained with the reference medicine in terms of efficacy and safety to demonstrate similarity.

Biosimilar medicines are approved in an abbreviated procedure that guarantees similarity, but not equivalence, with the pre-existing reference medicine. In this context, comparability studies make it possible to generate the evidence required to substantiate the similarity in the areas of quality, safety, and efficacy, thus guaranteeing that the efficacy and safety identified for the reference medicine are recognised for the biosimilar medicine.

However, how is similarity addressed? How is the absence of significant clinical differences reported? High similarity and no significant clinical differences from the reference product are the two key requirements for biosimilar products. Recent reviews have addressed the pivotal regulatory considerations to support similarity assessment so that these questions are answered and clarified. In this sense, the assessment of similarity first involves a structural and functional characterisation/evaluation, followed by non-clinical and clinical studies, adopting an adapted approach throughout the biosimilar development process. It is important to emphasise that the aim of comparative clinical studies is not to re-establish efficacy and safety but to identify any differences considered clinically significant between the proposed biosimilar and the reference medicine. Nevertheless, the approval of biosimilar medicines is highly regulated, so differences between the biosimilar medicine and its reference medicine must be supported by strong scientific evidence that these differences are not clinically significant [8,9]. The World Health Organization also presents a guideline on the evaluation of biosimilars, in order to outline the globally accepted principles for licensing biosimilar medicines, i.e., medicines that are highly similar to reference biological medicines in terms of quality, safety, and efficacy [10].

## 3. Development of Biosimilar Medicines

In terms of development, biosimilars entail costs approximately one hundred times higher than those associated with generic drugs, with a development timeline ranging between seven and eight years, as opposed to the two to four years typically required for generic drugs [11]. The basis of the development of a biosimilar drug is thus an extensive structural and functional characterisation as well as the comparison of the biosimilar drug with the reference drug.

The development of a biosimilar medicine starts with defining the fingerprint of the reference medicine and its attributes, establishing the boundaries of potential variability for the biosimilar. Since the manufacturing process of the reference molecule remains undisclosed, a novel process must be devised to ensure alignment with the fingerprint. Such a process requires that the cell culture and purification process conditions are continuously adjusted, exploring new cell lines throughout the development until the highest similarity is achieved. Progress towards clinical trials for the potential biosimilar hinges on achieving full characterization of the molecule, well-defined processes, and confirmed molecular similarity [7,12,13]. The development process for biosimilars is complex, encompassing various stages such as cell line selection, cultivation, production, isolation, purification, and, ultimately, formulation, filling, and finishing.

The first decision in the development of a biosimilar is about the cell line creation (Figure 1), being such an important decision for the glycosylation patterns and the determination of the final profile of the biosimilar so that the protein of interest is expressed. The relevant gene is then cloned into a complementary DNA vector, which could be derived from human or microbial cell lines. Generally, the industry opts for Chinese hamster ovary cells for expression cells due to their consistent folding, ability to thrive in suspension, high yield, and stability against changes in pH and oxygen levels [14,15,16,17]. Following this step is clone selection, aiming to identify clones that closely mirror the desired product fingerprint. It is crucial to note that no two biologics are identical as each product is manufactured using a unique cell line from the producer [18]. It is important to mention that there are no two identical biologics because each product is produced using an exclusive cell line from the manufacturer [19,20].

Moving towards the second step in the development of a biosimilar, cultivation and production (Figure 2), the cell line that will produce the protein and originate the biosimilar medicine is expanded in a fermentation medium to obtain a batch of master cells, and these cells are then cultivated and grown in large-scale bioreactors [14,15,22,23]. First, thawing of the cells occurs, followed by inoculation under agitation to increase cell density. To ensure cell viability, the cells are maintained in a growth medium fortified with nutrients and supplements [24]. Finally, it is also important to mention that throughout this process, parameters such as the amount of oxygen, lactate production, temperature, pH, and osmolarity are controlled [20,25].

The third step consists of isolation and purification (Figure 3). It is intended to recover the target protein, discarding unwanted impurities whether they are viruses, proteins, host cell DNA, aggregates, or endotoxins [26]. At this stage, the protein of the biosimilar is excreted in the cell culture fluid by the mammalian cell, and it is necessary to recover it by removing cellular debris by centrifugation and filtration. In addition, cellular metabolites are also eliminated so that they do not generate immunogenicity [24]. Still, within the scope of purification, it is important to highlight that such a process depends on column chromatography and filtration to remove unwanted impurities, using the biochemical properties that are similar to what is intended to be eliminated [20]. At this stage, the procedures are also constantly monitored, namely, pH, conductivity, and flow rate [20,27].

Finally, the last stage concerns the following topics: formulation, fill, and finish (Figure 4). The formulation of the product inevitably influences the mimicking of its degradation and the maximisation of its shelf life, involving the optimisation of the buffer solution conditions, such as pH, ionic strength, excipients, and stabilisers [24]. In this stage, the product is subject to specific conditions to assess its behaviour and stability, such as exposure to high levels of agitation or thermodynamic stress, such as high temperatures, freezing, or thawing for long periods. Regarding the final presentation of the drug, it can take three possible forms: liquid, frozen liquid, or lyophilized. The preferred option is typically the liquid form, although this choice depends on the stability of the product [7,20]. Freezing the product during storage may be necessary to mitigate chemical degradation. Opting for a lyophilized presentation entails additional costs and necessitates the use of diluents for reconstitution. Alternatively, if the frozen form is chosen, a cryoprotectant such as sucrose should be added to minimize cryoprecipitation or aggregation caused by cryoconcentration [20,25].

The manufacturer of the biosimilar medicine must develop the entire manufacturing process, from cell line selection to production. These activities are performed without full access to the development history of the reference medicine. Consequently, there exists significant potential for variations between an innovative biological medicine and a biosimilar medicine in terms of manufacturing processes. Therefore, any differences in the safety/efficacy profile should always be justified. It should also be noted that modifications for the purpose of improving efficacy are not expected to be accepted as distinct from the concept of a biobetter. The comparative studies between the biosimilar biological medicine and the reference medicine include the criteria of quality, safety, and efficacy, all aimed at demonstrating compliance with approved regulations [28].

Pharmaceutical quality then focuses on characterisation at the structural level, through physicochemical properties, which encompasses control and production standards and product preparation and processing. In addition, purity must also be controlled so that it does not exceed the established limit. Other data concern the biological activity, the excipients and starting materials, the dosage and formulation, the control of the manufacturing process, and the actual stability of the active substance and the final product [1].

### 3.1. Immunogenicity

Concerning immunogenicity, proteins inherently possess the capacity to elicit an undesired immune response. In rare instances, this reaction can lead to severe adverse effects or even diminish efficacy, although such occurrences are infrequent. Immunogenicity is influenced by various factors, including the drug characteristics, external factors associated with treatment, and individual patient- or disease-related factors. Quality issues, such as alterations in production methods, formulations, or packaging, can impact immunogenicity by affecting the drug properties and impurity profile. However, it is crucial to ensure that these changes do not compromise the medicine’s efficacy and safety [29,30]. It is very unlikely for an adverse immune reaction to occur following the aforementioned modifications as comparability studies indicate no significant increase in impurities or aggregates despite these alterations. Regulatory agencies actively monitor the immunogenicity of biosimilar drugs, employing comparability studies across different batches and utilizing physicochemical and structural analyses alongside functional in vitro assays [31,32].

The immunogenicity data required for the approval of a biological medicinal product for approval purposes, as highlighted by the EMA, include the incidence, titration, and persistence of antibodies against the biological medicinal product; neutralisation tests; clinical impact assessment; and measures to manage the risk of immunogenic potential. Nevertheless, these data always depend on the type of biological medicine and its intended use and the characteristics of the product itself, among others [1,33].

### 3.2. Extrapolation

Extrapolation represents a well-established scientific concept, aiming to evaluate the extent to which a biosimilar closely resembles a reference medicinal product in terms of safety and efficacy within a particular therapeutic indication. This similarity allows for the extension of these findings to other approved indications for the reference product. In practical terms, this means that, in some situations, few clinical studies with biosimilars may be carried out since this extrapolation is always supported by scientific evidence from comparability studies. Extrapolation criteria typically include considerations such as the mechanism of action, the target study population, various clinical settings, safety profiles, and immunogenicity data [1,34].

Concerning the mechanism of action of the active substance, it should be mediated by the same receptor, and additional research may be necessary to demonstrate similarity in behaviour. Depending on the clinical context, variations in dosage, pharmacokinetics, or mode of action may arise, necessitating further studies as the data for one indication may not directly apply to another. Establishing a comparable safety profile for a specific indication is crucial at this stage before extrapolating safety data. Immunogenicity data, being a specific criterion, require thorough substantiation and add complexity to the process [1,35].

### 3.3. Safety

In addressing the safety of biosimilar medicinal products, a comprehensive approach is essential, from a sound regulatory framework, a risk management plan always in place, post-authorisation safety studies, and continuous safety monitoring. Given that many adverse drug reactions associated with biosimilars can be anticipated based on their pharmacological effects, efforts are made to collect reports of adverse reactions spontaneously. This includes the submission of periodic safety reports and the implementation of additional monitoring mechanisms to detect potential long-term adverse events or with long latency periods [1].

The EU has a consolidated system of monitoring, reporting, assessing, and preventing adverse drug reactions, so the authorities continuously assess the risk–benefit ratio of all medicines and take the necessary measures. When an undertaking applies for the marketing authorisation, it will submit a risk management plan including a pharmacovigilance plan as well as any risk minimisation measures to identify, characterise, and minimise risks. When talking about a risk management plan (RMP) for a biosimilar medicine, it will always be based on the knowledge and experience already gained with the reference medicine. For post-authorisation studies, it has been made possible to monitor known risks and detect adverse drug reactions that only arise after a large number of patients receive treatment over a substantial time period. In addition, it is noteworthy that toxicity studies are used to carry out this safety assessment of biological medicinal products. The main mechanisms of adverse effects of biological drugs are related to exaggerated pharmacology. This phenomenon, known as on-target toxicity, entails the occurrence of adverse pharmacological effects specifically at the intended target site. However, if comparable pharmacological activity has been established in vitro, there is no need to confirm these mechanistic properties in vivo [36].

Examples of unexpected toxicity are scarce. Often, functional differences between these antibodies have a structural basis rooted in variations in the amino acid sequence. These differences are not integrated into the development of similar biological medicines because the amino acid sequence should be the same as that of the reference medicine [37]. Thus, although unexpected toxicity can be found in pre-clinical animal studies during the development of new biological medicinal products, there is no evidence to suggest that such occurrences are associated with biosimilars [12].

### 3.4. Comparability

The fundamental principle of developing a biosimilar is based on the comparability exercise that is carried out between the reference biological medicine and the biosimilar medicine. For such a comparability exercise to be undertaken, comprehensive analyses of the proposed biosimilar and the reference product are required, using sensitive methods. The main objective is to demonstrate that the biosimilar and the reference product are similar at a finished medicinal product level [28].

In this context, several important pillars arise when comparing the reference medicine and the biosimilar medicine: target/quality by clinical trial design, quality (data set for stand-alone) biological and physicochemical comparability, comparative pre-clinical trials, comparative clinical trials, and risk management plan [11]. The comparability exercise is carried out in several steps, namely, quality comparability, non-clinical comparability, and clinical comparability.

In the first step, quality comparability, a full characterisation approach needs to be undertaken to compare the physicochemical and biological quality attributes, for instance, the purity of the potential biosimilar medicine. This is performed using a wide range of different state-of-the-art analytical tests, as no single method can characterise all aspects of a product. The development process is also to be modified if there are significant differences found in the analyses until the product generated has a profile that matches the reference product profile. Iterative adjustments are made at every phase of the development process to ensure that the ultimate biosimilar medicine meets the quality standards set by the EMA, aligning with all criteria stipulated for documentation submission, assessment, and marketing authorization.

In the second step, non-clinical comparability, non-clinical studies, sometimes called pre-clinical studies, need to be conducted for biosimilar medicines before any clinical trial begins. Generally, non-clinical data for a biosimilar are generated through an abbreviated testing programme or in vivo animal studies, as required by the EMA’s guidelines. On the other hand, non-clinical studies usually include repeated dose toxicity studies as well as pharmacokinetic and pharmacodynamic (PK/PD) studies along with local tolerance testing. For this type of study, similarity criteria should be pre-defined and scientifically justified in order to allow comparability of the support with the reference and detect potential differences between them.

In the third step, clinical comparability, clinical tests are also considered comparative in the case of the development of a biosimilar medicinal product. However, such clinical tests are not required to the same extent as would be necessary for a new active substance, taking into account the clinical experience gained from the use of the reference medicinal product. Therefore, it is crucial to consider the nature and characteristics of the medicinal product as well as the therapeutic indications. Another key aspect is to understand how comparable the profile of the biosimilar is to that of the reference product. The closer the biosimilar and reference profiles are and the greater the similarity demonstrated (through appropriate studies such as comparative quality, biological and receptor binding activity analyses, and animal testing), the more easily the clinical trial programme will be accepted by the regulatory authorities.

In summary, the clinical comparability exercise starts with pharmacokinetic and/or pharmacodynamic studies, and comparative clinical efficacy and safety trials may still follow. An important step is to document the side effects and to take into consideration the evaluation of immunogenicity for which comparable profiles for the biosimilar and the reference are also required for the clinical safety data assessment [38,39]. Apart from comparing the quality attributes of both medications, establishing the similarity in biological activity and safety of the biosimilar involves utilizing relevant and sensitive in vitro assays. These assays help identify significant differences in allosteric mechanisms between the two medications across all approved indications [40]. Table 2 outlines the comparability critical attributes and the way the similarity of the products is demonstrated [12,40].

In terms of protein structure, as a critical quality attribute, the similarity assessment includes a detailed analysis of conformational and post-translational modifications between a biosimilar and its reference medicine. This is fundamental in demonstrating comparability and ensuring the efficacy and safety of the biosimilar. These structural characterization studies involve sophisticated analytical techniques such as mass spectrometry, nuclear magnetic resonance spectroscopy, and X-ray crystallography to provide insights into protein folding, stability, and function. A thorough understanding of protein structure similarity is critical in establishing the interchangeability and substitutability of biosimilar medicines, ensuring patient safety and therapeutic efficacy [41].

### 3.5. Interchangeability

The term interchangeability emerged in the context of the possibility of exchanging a reference biological medicine for its biosimilar, or a biosimilar for another biosimilar, hoping to obtain the same therapeutic effect produced by the option initially used [1,42].

This exchange can occur in two distinct ways: switch or automatic substitution. As for the “switch” option, this occurs when the prescribing physician decides to exchange one medicine for another that has the same therapeutic purpose. Automatic substitution, on the other hand, refers to a practice carried out at the pharmacy level without consulting the prescribing physician, where the dispensing of the prescribed drug is changed by another equivalent and interchangeable drug [1,43].

These issues have been the subject of some controversy, despite the existence of several studies that prove the safety and effectiveness of this practice [44].

In terms of pharmacovigilance, the traceability of the biological medicine involved in the potential adverse reaction is very important, so the same brand of medicine should be kept for as long as the reaction is expected to occur. The change between biosimilar biological medicines should respect a minimum period of time that safeguards their traceability. This information can be found in the National Medicines Formulary and, when this is not mentioned, that period should not be less than 6 months. Switching between different brands of the same biological medicine must be articulated with the clinical services involved, with respect to the precautionary principle and in accordance with the therapeutic indications for each situation. This position is reviewed whenever applicable scientific evidence becomes available.

Therefore, what is the EMA’s position on the issue of switching the biosimilar as a reference medicine? Previously, the EMA positioning was directed to the independent deliberation by each member state.

In Portugal, the decision regarding therapeutic switching is made by INFARMED I.P. in conjunction with the National Pharmacy and Therapeutics Commission. They stipulate that the switch should occur only after the specified minimum duration to guarantee medicine traceability. However, automatic substitution is not currently permitted in Portugal [1,45,46]. Note, however, that in September 2022, as reported on the EMA’s website, the EMA and the Heads of Medicines Agencies (HMA) released a joint statement affirming the interchangeability of biosimilar medicines approved in the EU with their reference medicine or an equivalent biosimilar. This unified stance serves to standardize the approach across the EU [47].

## 4. Regulatory Framework for the Development of Biosimilars

The marketing authorisation for biosimilar medicinal products, via the EMA, is obtained through a centralised procedure, in which the Agency evaluates the medicinal products for the purpose of authorising their marketing. Comparative characteristics studies evaluate the composition, physical properties, protein structure, purity, isoforms, or impurities that derive from the product, as well as biological activity. The production process between the reference biological drug and the biosimilar may, however, present differences. Concerning the regulatory framework for the development of biosimilars, the perspective of the approval process via the EMA and FDA as well as other regulatory agencies may differ in certain aspects [6,48].

### 4.1. European Medicines Agency

All biotechnologically produced medicinal products are approved in the EU by the EMA through a centralised procedure. There are certainly some biosimilars, such as low-molecular-weight heparins obtained from the swine intestinal mucosa, which are approved at the national level; however, mostly, and since they resort to biotechnology for production, they are approved by the centralized procedure.

First, the entity submits the marketing authorisation application to the EMA. The data are then evaluated by the scientific committees, the Committee for Medicinal Products for Human Use (CHMP) and the Pharmacovigilance Risk Assessment Committee (PRAC), as well as by experts in biological medicinal products and biosimilar specialists. Subsequently, this analysis gives rise to a scientific opinion sent to the European Commission (EC). If applicable, it is for the EC to grant marketing authorisation in the EU and it is valid in all EU member states [1].

The EMA was the first organisation to delineate a specific regulatory route for biosimilar medicines; this happened in October 2005. The EMA guidelines state that the approval of biosimilars is based on comparability studies, through the characterisation of the protein structure of the biosimilar, as well as its efficacy, safety, and immunogenicity. In addition to these aspects, it is considered essential to carry out in vitro tests, impurity profile analyses, and pharmacokinetic and pharmacodynamic studies, as well as pharmacovigilance monitoring after its approval on the market [49].

There are several important requirements according to the Agency, including the importance of demonstrating robustness and consistency in its manufacturing process, as well as details for additional non-clinical and clinical studies to attest to the comparability.

A key point highlighted by the EMA is that the clinical benefit has already been proven by the innovative product; so, the goal of a biosimilar is to demonstrate similarity to the innovative product, in terms of strength, active substance, route of administration, and pharmaceutical form, and not the clinical benefit itself.

In order to compile all legislation concerning biological medicines, the EMA has adopted several more individualised guidelines for each type of biosimilar medicine, as presented in Table 3. The general guidelines include the Guideline on similar biological medicinal products, the Guideline on similar biological medicinal products containing biotechnology-derived proteins as active substances: quality issues, and the Guideline on similar biological medicinal products containing biotechnology-derived proteins as active substances: clinical and non-clinical issues [4,28,31,50,51]. In addition, the EMA has also developed more specific guidelines: the Guideline on biosimilar medicines containing recombinant granulocyte colony-stimulating factor, Guideline on clinical and non-clinical development of similar medicines containing low-molecular-weight heparins, Guideline on clinical and non-clinical development of similar medicines containing recombinant human insulin and insulin analogue, Guideline on clinical and non-clinical development of similar medicines containing beta interferon, Guideline on clinical and non-clinical development of similar medicines containing monoclonal antibodies: ethical and non-clinical issues, Guideline on clinical and non-clinical development of similar medicines containing recombinant erythropoietin, Guideline on clinical and non-clinical development of similar medicines containing recombinant follicle-stimulating hormone, and Guideline on clinical and non-clinical development of similar medicines containing somatropin [4,28,31,50,51]. Finally, two other relevant guidelines on the comparability of biotechnology-derived medicinal products after a change in the manufacturing process exist: clinical and non-clinical issues and another one on ICH Q5E biological/biotechnology products subject to changes in their manufacturing processes: comparability of biological/biotechnology products [4,28,31,50,51].

### 4.2. Food and Drug Administration

In the United States, the creation of guidelines for biosimilars did not happen as early as in the EU. The FDA has only recently started formulating distinct guidelines for the approval of biosimilar drugs. At the level of important factors to consider at the time of the FDA’s approval process, the need for robustness in the manufacturing process, the similarity of the protein structure, the extent of understanding of the mechanism of action, appropriate pharmacodynamics trials, comparative pharmacokinetic data, immunogenicity data, and clinical data regarding the innovative product are highlighted [49].

In the initial phase, the biosimilar medicinal product entity presents a series of data comparing the candidate medicinal product with the reference medicinal product in order to demonstrate similarity. First, the data are evaluated, starting with a detailed analytical characterisation, at the structural and functional level, as well as a comparison, moving to animal studies, and, if necessary, comparative clinical studies [18].

The submission of a biosimilar medicinal product will include some data to demonstrate such similarity with the reference medicinal product, from analytical studies demonstrating that the biosimilar medicinal product is highly similar to the reference medicinal product to toxicity assessment studies. Another aspect refers to a clinical study to demonstrate the safety, purity, and potency of the proposed biosimilar product in one or more of the indications for which the reference product is licensed. This typically includes the evaluation of immunogenicity, pharmacokinetics (PK), and, in some cases, pharmacodynamics (PD) and may also include a comparative clinical study [53].

More practically, the application for marketing authorisation is made to the FDA and submitted through section 351(k) of the Public Health Service Act (PHS). This section allows the biosimilar medicinal product to be approved for all therapeutic indications of the reference medicinal product. This happens several times when discussing the speed of the drug approval process in the United States. This situation is due to the fact that, in 2010, a change was made to the Biologics Price Competition and Innovation Act, enabling the existence of a shortened route for the regulation of biosimilar drugs in the USA. However, it is important to point out that such a change does not diminish the required standards of safety, purity, and efficacy.

As with the EMA, the FDA requires a rigid, sequential approach with rigorous comparability tests, from physical-chemical, analytical, functional, and non-clinical and clinical assessments.

In 2014, the Purple Book was published by the FDA, where reference drugs and corresponding biosimilars are found, presenting the medicines authorised for commercialisation in the USA [54,55]. In 2015, the FDA published three guidelines, granting industry guidelines for the development of biosimilar drugs. These include scientific assessments in the demonstration of similarity, considerations about the quality for the demonstration of similarity with the reference medicinal product, and a set of questions and answers on the evaluation of the drugs in question [16]. In 2018, the FDA’s Biosimilar Action Plan (BAP) was published, providing information regarding actions to stimulate the development of biosimilars. Briefly, the BAP analyses the phases so that development and approval are more effective, clarifying regulation and developing support materials for healthcare professionals and patients to demystify the issue and increase confidence. Nevertheless, the BAP does not address issues related to improvements in pharmacovigilance [54,56].

Table 4 summarises the key criteria for developing a biosimilar medicine through FDA legislation. The FDA’s determination of similarity is based on the totality of evidence provided in the marketing application for FDA review. The data set in the marketing application includes extensive analytical comparison to show that the proposed biosimilar and the reference products are extremely similar in structure and function. Animal, human pharmacological, immunological, and other data are added as needed to the analytical data in a stepwise manner to provide the necessary information with the ultimate goal of demonstrating similarity [13].

### 4.3. Other Regulatory Authorities

In addition to the EMA and FDA, there are other regulatory agencies also responsible for the evaluation and monitoring of medicines submitted for marketing authorisation.

In Japan, the Pharmaceuticals and Medical Devices Agency (PMDA), in particular, the ‘Office of Cellular and Tissue-Based Products’, stands out as the organisation responsible for the assessment, regulation, and authorisation of biosimilar medicines, dealing with a wide range of activities, from approval reviews to post-authorisation surveillance. The Japanese regulation is based on European legislation and, therefore, bears a strong resemblance to the binding model of the EMA. The PMDA also highlights the need to conduct clinical trials in order to demonstrate clinical comparability between the biosimilar medicine and the reference medicine. Nevertheless, sometimes conducting comparative studies on pharmacokinetics and pharmacodynamics can be considered sufficient to demonstrate clinical comparability and no further clinical trials are required [57,58]. Another debated point from Japan concerns extrapolation, and this can be carried out for all therapeutic indications of the reference medicinal product, provided that equivalent therapeutic effects and clinical comparability are demonstrated [59,60].

Regarding the Middle East market, the emergence of biosimilar medicines has allowed such medicines to reach other countries characterised by a high percentage of poverty, avoiding the poor-quality imitations that are so typical of these regions [35]. The Central Drugs Standard Control Organisation is the organisation responsible for medicines in India, and its mission is to ensure that all procedures are reliably controlled. Under Indian law, a biosimilar medicine can be approved faster if its reference medicine is approved in the EU or the USA. For example, a biosimilar can be approved with phase III bioequivalence studies with about 100 patients, while the EMA requires much more, up to 500. A major peculiarity of the Indian legislation is the lack of market exclusivity for products that are approved for the first time. In these cases, this exclusivity relates only to the patent plus the fact that the patent is less developed and, therefore, considered a lower barrier [59,61].

In Canada, the drug approval process is handled by the Biologics and Genetic Therapies Directorate (BGTD) of the Health Products and Food Branch (HPFB) of Health Canada. Currently, there are specific guidelines for biosimilar medicines prepared by Health Canada to have more authoritative legislation. The first biosimilar drug approved in Canada was a growth hormone, through Sandoz, called OMNITROPE, in 2009 [62,63].

In South America, regulatory guidelines follow the EMA’s and FDA’s models, with some particularities certainly, as will be shown in the following paragraphs.

For the registration of biosimilars in Venezuela, the National Institute of Hygiene Rafael Rangel is used; it is based on the generation of information on biosimilarity in terms of the quality, safety, and efficacy of the reference medicine. It should be noted that extrapolation to use indications is not allowed, so individualised/separate information is generated [48,64].

The Instituto Nacional de Vigilancia de Medicamentos y Alimentos is the regulatory authority in Colombia and it is responsible for the inspection and supervision of the marketing and development of health products, as well as the identification and evaluation of sanitary standards and procedures, among others [46].

In Brazil, according to Brazilian legislation and the National Health Surveillance Agency, the registration of biosimilars is performed after the submission of comparative studies between the biosimilar and the reference medicine that confirm the quality, efficacy, and safety requirements. When a biosimilar drug is in the approval stage, extensive pre-clinical documentation is required, which should show such points of similarity, ensuring its criteria and attributes [45,65].

In Chile, the responsible regulatory agency is el Departamento Agencia Nacional de Medicamentos, and, concerning biosimilars, this is still a developing country in terms of the regulation necessary for them [49].

Argentina’s regulatory agency is called la Administración Nacional de Medicamentos, Alimentos y Tecnología Medica and it is governed by the guidelines of the EU model [66].

In China, the Center for Drug and Food Evaluation of the China Drug Administration issued guidelines on “similar biologics” on 28 February 2015. According to the Chinese guidelines, a similar biologic is a “therapeutic biologic” similar to an authorised reference product in terms of quality, safety, and efficacy. Regarding clinical trials, the Chinese guidelines suggest that clinical trials for “similar biologics” should start with PK and PD studies based on which clinical efficacy and safety studies should be conducted. Comparability between the similar biological product and the reference product may be performed for quality assessment and non-clinical studies based on PK, PD, and PK/PD clinical studies. The Chinese guidelines suggest that the assessment of clinical immunogenicity should be based on the results of a non-clinical immunogenicity assessment. Where such studies suggest the similarity of the similar biological product to the reference product, then only limited clinical evaluation is required. The approval of “similar biological products” has not been assigned to a separate abbreviated approval route. The approval pathways are subject to the same pathways that apply to innovative biological products. In addition, for the reference product, the Chinese authorities require the product to be approved in China. Reference products approved by foreign regulatory authorities and not approved in China are not acceptable as reference products [53,54].

The Ministry of Food and Drug Safety (MFDS) of the government of South Korea, through its National Institute for Food and Drug Safety Evaluation, seeks to scientifically evaluate drugs developed by South Korea’s pharmaceutical industries. Of note, the national guidelines were based on those of the World Health Organization, the EU, and Japan [52].

Some countries in Asia do not have their own agency that focuses on the evaluation of biosimilar medicines. Hence, the Association of Southeast Asian Nations (ASEAN) emerged; it represents 10 member states in the ASEAN region including Indonesia, Malaysia, Thailand, the Philippines, Singapore, Darussalam, Vietnam, Laos, Myanmar, and Cambodia. In this sense, it is through this Association that many decisions of scientific, political, and economic issues, among others, are made in order to have an important role in this region [67].

## 5. Pharmaceutical Market for Biosimilars

Based on a market research report, the biosimilar medicines market demonstrates continuous growth. This trend persists as numerous biological molecules enter the market, introducing novel therapeutic options across various therapeutic areas for patients [68].

### 5.1. Remicade^®^, a Reference Medicine

In order to understand the regulatory framework of a biosimilar medicine, a more practical study of Remicade^®^ and its biosimilars was undertaken. The approval process described was centralised by the EMA.

Remicade^®^ is an anti-inflammatory drug, usually used when other medicines or treatments have failed in adults with the following diseases: rheumatoid arthritis, Crohn’s disease, ulcerative colitis, ankylosing spondylitis, psoriatic arthritis, and psoriasis [69].

In the case of rheumatoid arthritis, it is an immune system disease that causes inflammation of the joints, and Remicade^®^ is given together with another medicine, Methodrex. Crohn’s disease is a disease of inflammation of the digestive tract, which can lead to the formation of fistulas. Ulcerative colitis and ankylosing spondylitis are diseases that cause inflammation, under the effect of ulcers in the lining of the intestine and pain in the joints of the spine, respectively. Finally, psoriatic arthritis and psoriasis cause the appearance of desquamative red plaques on the skin, the first of which simultaneously causes inflammation of the joints.

Regarding how Remicade^®^ is used, this is the preparation of a solution for infusion in the form of powder. At the time of treatment, all patients are monitored for possible reactions, and it is therefore strictly crucial that it be administered with the supervision of a specialist doctor.

Remicade^®^ may also be used in younger patients, aged 6 to 17 years, who have Crohn’s disease or severe ulcerative colitis and who have not been responsive to other medicinal products/treatments or even are unable to receive such therapeutic strategies [69,70]. Nevertheless, there are few biosimilars with therapeutic indications approved for paediatrics. This is because we are talking about biological medicines with larger and more complex molecules than a chemically synthesised medicine. However, the administration processes are more invasive, using the parenteral route, which means an added difficulty for children. In this sense, this point is still considered a delicate issue, and the regulatory agencies themselves do not always agree in this area [71,72].

The active substance in Remicade^®^ is infliximab, which is a monoclonal antibody, a type of protein that is designed to recognize and bind to a specific structure antigen [69].

In practical terms, infliximab binds to a chemical messenger, tumour necrosis factor-alpha, and, once it is involved in the inflammation process, acts directly in this direction. Thus, it will promote TNF-α blockade, reducing inflammation and other symptoms associated with the disease in question [73,74].

Remicade^®^ is a reference medicine with the following characteristics: 100 mg concentrate powder for infusion solution. Each vial contains 100 mg of infliximab. The mode of administration is intravenously over 2 h. Remicade^®^ belongs to the pharmacotherapeutic group immunosuppressants, as inhibitors of tumour necrosis factor-alpha (TNF-α) [75].

Regarding its mechanism of action, the active substance (infliximab) is a man-murine chimeric monoclonal antibody that binds with a high affinity to both soluble and transmembrane forms of TNF-α but not a lymphotoxin (TNF-β). When administered at the recommended doses and concentrations, it is an effective, safe, and generally well-tolerated drug. Its mechanism of action involves the neutralization of the biological activity of TNF-α through the binding with high affinity to the soluble and transmembrane forms of TNF-α and inhibits the binding of TNF-α with its receptors. This neutralization leads to an overall reduction in inflammation [74].

### 5.2. Case Study: A Reference Medicine (Remicade^®^) vs. a Biosimilar Medicine (Inflectra)

The Remicade^®^ approval process, as described in Figure 5, began on 27 March 1998, and the first evaluation report was released to all members of the Committee for Medicinal Products for Human Use (CHMP) on 8 June 1998. Subsequently, some questions were raised, and a consolidated list of questions was transmitted to the company on 30 November 1998. The evaluation report with the answers to these questions was then released by the rapporteur within the CHMP on 25 January 1999. On 24 February 1999, the CHMP discussed this same report and convened a group of experts to reflect on some specific clinical points, including safety, post-marketing, and efficacy studies. In this context, a list of quality aspects to be taken into account in this medicinal product was also recommended. Finally, on 19 May 1999, the CHMP, based on the data submitted as well as the scientific discussion within the Commission, issued a positive opinion for the granting of a marketing authorisation to Remicade^®^. It should also be reported that such opinions were also forwarded to the EC to follow up on this decision [76].

Concerning the reference medicine Remicade^®^, four biosimilar medicinal products were approved for use in the European Area. The four biosimilar drugs (Inflectra, Remsima, Flixabi, Zessly) are anti-inflammatory drugs whose active substance is infliximab. In this sense, the four medicinal products are similar to a biological medicine, called a reference medicine, already authorised in the EU: Remicade^®^.

Such medications are used in the treatment of adults with the following diseases: rheumatoid arthritis, Crohn’s disease, ulcerative colitis, ankylosing spondylitis, psoriatic arthritis, and psoriasis. Furthermore, they can be used in children between 5 and 17 years of age who have Crohn’s disease or severe ulcerative colitis and have been first submitted to other treatments or medications without success. Its mode of administration is via infusion form, into a vein for 2 h. With regard to the approval process, it has already differed among the four medicinal products, although it is similar [77,78,79,80].

#### 5.2.1. A European Approach to Biosimilars’ Approval Process

Regarding the approval process of INFLECTRA, through the EMA, this was through a centralized procedure, as well as the others mentioned previously.

The procedure began on 14 July 2012, followed by a meeting between 16, as fully described in Figure 5, and 19 July 2012, in which the CHMP drew up a consolidated list of questions to send to the applicant, which was sent the following day to the applicant. On 16 November, the applicant submitted the answers to the consolidated list of questions. Following this, the rapporteurs released the joint evaluation report on such responses to all CHMP members on 9 January 2013. During the meeting from 7 to 10 January 2013, the Pharmacovigilance Risk Assessment Committee (PRAC) adopted the PRAC Council’s Risk Management Plan. Still within the CHMP, a list of outstanding issues was issued in writing or orally. On 29 April 2013, the applicant then submitted the answers to outstanding questions. To this end, the PRAC again adopted the Council’s RMP, and the rapporteurs released the joint evaluation report on such responses to all CHMP members on 24 May 2013. The applicant then orally stated the outstanding issues raised, between 27 and 30 May, and the CHMP agreed to a second list of outstanding issues to be addressed in writing by the applicant. After the applicant submitted the replies to the second list and the PRAC had gathered, the rapporteurs reissued the joint evaluation report and, on 27 June 2013, the CHMP, taking into account all the data collected and the scientific discussion generated, issued a positive opinion to grant a marketing authorisation. Figure 5 shows the approval process for Remicade^®^ and its biosimilar Inflectra, via the EMA’s centralised pathway [77].

#### 5.2.2. A United States of America Approach to Biosimilars’ Approval Process

It is also important to mention that Inflectra was the second biosimilar approved by the FDA and the first Remicade^®^ biosimilar also approved for marketing in the United States. According to the FDA, the approval of Inflectra was based on a review of evidence including structural and functional characterisation, animal study data, pharmacokinetic and pharmacodynamic data, clinical immunogenicity data, and other clinical safety and efficacy data that demonstrated that Inflectra is biosimilar to Remicade^®^ [77].

## 6. A Critical Appraisal of Biosimilars

Reflection on biosimilar medicines becomes fundamental to bringing together ideas, discussing issues, and, essentially, reflecting on their benefits, risks, regulatory particularities, and future perspectives.

### 6.1. Challenges on Biosimilar Medicines

Currently, at the scientific level, one of the great challenges of efficacy and safety of biosimilar medicines lies in the difficulty of demystifying the concepts of biosimilar, generic, and biobetter. The three designations differ since generic medicines are equivalent to reference medicines and biosimilar medicines are highly similar to existing biological medicines. Regarding biobetter medicines, while the main objective of developing a biosimilar medicine is to clearly and evidently demonstrate high similarity to the reference medicine, the objective of a biobetter medicine is not to show similarity but to reveal beneficial clinical superiority for the patient over the original biological medicine. The studies already carried out and the knowledge about these medicines make it possible to educate society in this regard and make it more receptive to this development in science. It is therefore important to communicate, inform, and clarify to both patients and healthcare professionals that these therapeutic strategies are available at a lower cost but with the same quality, efficacy, and safety. The biggest difficulty/challenge in researching a biosimilar will always be to prove similarity and not non-inferiority to the reference drug.

Another challenge is the misinformation regarding these medicines, which promotes ignorance and naturally generates distrust regarding the quality, safety, and efficacy of biosimilar medicines. Nowadays, in the field of research, it is possible to state that during the whole process of development and production of a biosimilar, the quality, safety, and efficacy profiles are proven through clinical trials that assess the pharmacokinetics, efficacy, safety, and immunogenicity of the biosimilar compared to the reference one. In this way, it is proven, through tests that demonstrate that the properties of the product, which may affect its immunogenicity, have not been altered.

Bearing in mind that, although the market for biosimilar medicines is very complex, specific, and large, it also entails high costs. The cost required to develop a biosimilar medicine, in itself, is a challenge, considering the high-tech equipment and manufacturing material needed, making it very expensive throughout the process, including all the clinical trials needed to prove its efficacy, also highlighting the importance of investment in human resources, since it is essential to involve specialized personnel in different fields of science to ensure the proper development of this type of medicine.

One of the most referred to and extremely important challenges in the use of biosimilars refers to regulation, which allows professionals to have safety and trust in their administration. In general, the existing legislation can still be considered quite recent, causing a certain instability and insecurity in its implementation in some countries. Nevertheless, the marketing of biosimilar medicines has more than 10 years of experience in the EU, and, during this time, biosimilars have maintained safety, efficacy, and quality throughout the process.

In reality, the competent authority for the authorisation of biosimilar medicinal products is the same as for biological reference medicinal products and mandates rigour and compliance with the required criteria for safety, efficacy, and quality. Regarding pharmacovigilance, biosimilar medicines, as well as reference biological medicines, are identified as a group of medicines that require additional monitoring. A risk management plan presents the identified risks, minimisation measures, and strategies to collect immunogenicity data.

A much-discussed aspect among scientists refers to interchangeability and substitution, which must be analysed case by case; it is the obligation of health professionals to be informed about the benefits and risks that such a change may bring to the patient. To this end, the study of the patient and the clinical analysis are essential for adequate and effective medical advice in order to consider the best therapeutic option for the patient in question, according to the stage of the disease.

Yet another challenge in this area is to prove that the extrapolation of data is preserved, which is easily corroborated with all the justification submitted to the EMA and additional clinical trials requested, as well as the inclusion of additional safety measures for monitoring these medicines to ensure their safety.

Understanding the implications of biosimilars and their dissemination is crucial for investment decisions. Deloitte has conducted a study on market entry strategies for biosimilar medicines to explore the investment opportunities available to pharmaceutical companies in this field. Regarding the goals and objectives defined, one should reflect on the real value of the business, keeping in mind the global vision of biosimilars, including the financial and non-financial goals. Next, it is crucial to understand which countries and regions can be considered and the potentials for this market segment and also substantiate which are the most attractive therapeutic areas, specifying what the missing needs in this area are. Bearing in mind the possibility of the existence or emergence of competing biosimilars and new commercial partners, it is important to reflect on their influence on the competitive market and, consequently, the prices charged, which would influence the cost and profit ratio.

### 6.2. Role of Biosimilar Medicines

The use of biosimilars in therapeutics in the socio-economic context we are living in can be considered as a solution for many therapeutic strategies. By reducing the costs associated with reference biological medicines, biosimilars address the pressing needs of the population. Moreover, studies supporting their efficacy confirm their suitability for meeting both short- and medium-term healthcare requirements. In the case of treatments reserved for the last therapeutic stages, in a more advanced phase of the disease, or for more serious patients, we can state that, through the administration of biosimilars, a greater number of patients would be able to access the medicines and more quickly. In addition, through cost reduction with biosimilars, the additional budget could be directed to areas of innovative and/or underserved therapies, thus contributing to the very high expenses or the inadequacy of the budget.

Biosimilar medicines today are a reality. They represent a complex and innovative world but with great potential to acquire a promising and prominent role in society. It is expected that investment will be made in the improvement of legislation regarding biosimilars so that the desirable and necessary harmonisation of guidelines becomes a practical and not just a theoretical reality.

Hitherto, biosimilar medicines have demonstrated their importance and efficacy in the fight against diseases and mainly for the well-being of individuals who for various reasons have developed a disease. This contribution is poised to significantly enhance the quality of life for people worldwide. As patents on biological medicines expire, the potential for biosimilar medicines to become a more equitable and sustainable state-of-the-art treatment will become increasingly evident and achievable.

## 7. Conclusions

This review aimed to reflect on and understand the procedures regarding the regulatory framework for biosimilar medicines in the world, bearing in mind the definition in the Portuguese legislation, which defines medicine as ‘any substance or combination of substances presented as having properties for treating or preventing disease in human beings or its symptoms or which may be used in or administered to human beings to establish a medical diagnosis or, by exerting a pharmacological, immunological or metabolic action, to restoring, correcting or modifying physiological functions’. As there are scientifically proven biosimilar medicines within this definition, it is important to reiterate that, when the patient benefits from their administration, clinical practice should be patient focused.

Moreover, as biosimilar medicines are a clinical alternative to biological medicines, their emergence follows the fact that biological medicines are associated with very high costs, preventing many patients with different chronic diseases from seeing their therapeutic strategy as something merely unaffordable. Currently, biosimilar medicines, characterised by their high similarity to reference biological drugs, represent a pivotal advancement in cost-effective pharmaceutical solutions, significantly enhancing accessibility to such treatments.

In this line of research, the EU, through the quality of the EMA, has been standing out for its intervention in the field of biosimilars, having already more than 10 years of proven clinical experience. Thus, it can be said that the Agency has been a pioneer in the approval of biosimilar medicines through the drafting of specific quality, safety, and efficacy guidelines for biosimilars. According to current data by regulatory agencies, over 86 biosimilars have been approved for use in Europe since 2006, compared to over 42 biosimilars approved for marketing in the United States. Both Agencies seek to excel in the quality, safety, and efficacy of their medicines by requiring several tests, in particular, comparative studies between the biosimilar medicine and its reference medicine in order to demonstrate the comparability between the medicines and ensure their high similarity.

In this sense, a critical appraisal of the regulatory framework for biosimilars was considered, focusing on four biosimilars of Remicade^®^—Inflectra, Remsima, Flixabi, and Zessly—that have been approved by the EMA. Throughout the approval process of these biosimilars, each critical step was listed so that the parameters requested by the Agency were met. To ensure compliance with such requirements, the actions of the Committee for Medicinal Products for Human Use as well as the presence of the advisory body of the Pharmacovigilance Risk Assessment Committee, are essential for the EMA to give its positive opinion regarding the introduction of a new medicine on the market. Throughout scientific research, issues such as demonstrating biosimilarity, extrapolation, safety monitoring, interchangeability, switching, and substitution have become strategies inherent in the world of biosimilars that must be studied continuously. Despite the evolution and proof of their efficacy, there is still much to be done, as ignorance of them is still very prevalent.

Considering the accomplished regulatory perspective on biosimilar medicines, the key to the success of these medicines lies in an even more rigorous, demanding, and universal regulation, as well as a greater knowledge, acceptance, and awareness from health professionals, since they are the ones who play the role of credible informers both for patients, through recommendation and prescription, and for potential investors in their production, in order to ensure the safety and efficacy of more accessible medicines. Additionally, when biosimilar medicines are used, these medicines bring benefits to the national health system, companies, and patients. Hence, biosimilar medicines can enable more patients to be treated with biological strategies at an earlier stage of the disease, which is more affordable. During this research, it was then possible to conclude that, even though the topic is only a few decades old, it is already well grounded and documented, with reliable and sound studies on the topic.

## Figures and Tables

**Figure 1 pharmaceutics-16-00321-f001:**
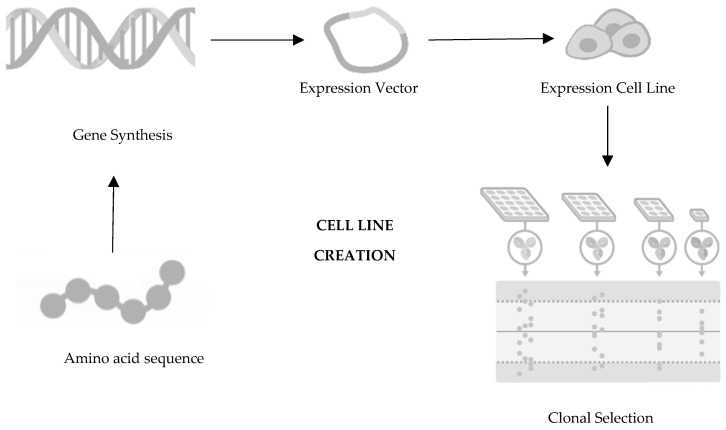
First step of development of a biosimilar: cell line creation. Adapted from [21].

**Figure 2 pharmaceutics-16-00321-f002:**
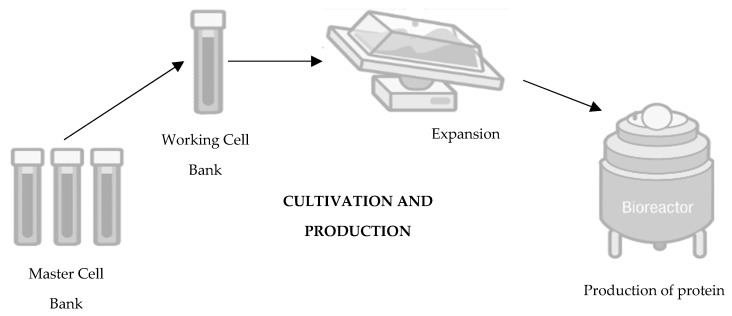
Second step of development of a biosimilar: cultivation and production. Adapted from [21].

**Figure 3 pharmaceutics-16-00321-f003:**
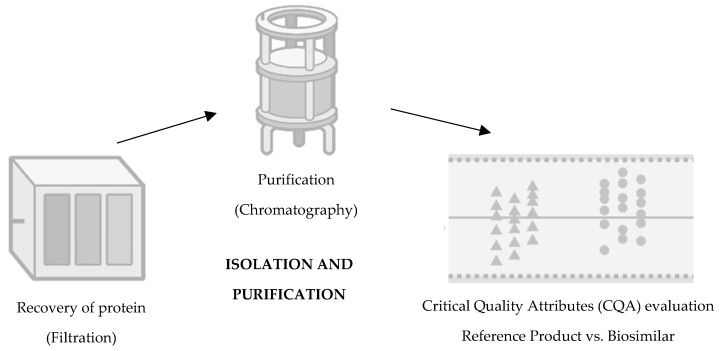
Third step of development of a biosimilar: isolation and purification. Adapted from [21].

**Figure 4 pharmaceutics-16-00321-f004:**
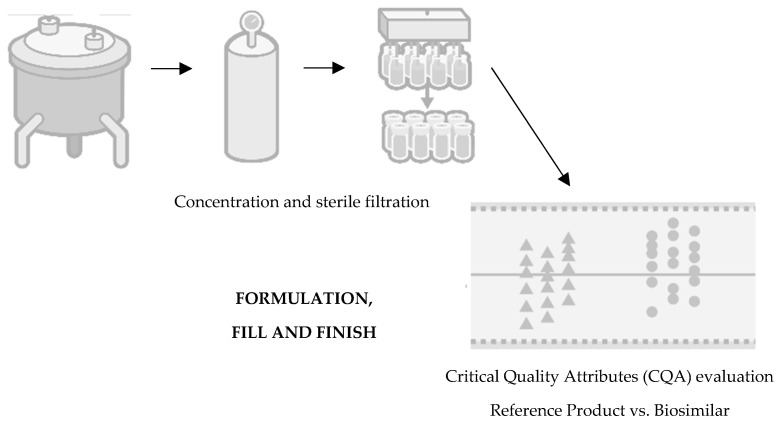
Fourth step of development of a biosimilar: formulation, fill, and finish. Adapted from [21].

**Figure 5 pharmaceutics-16-00321-f005:**
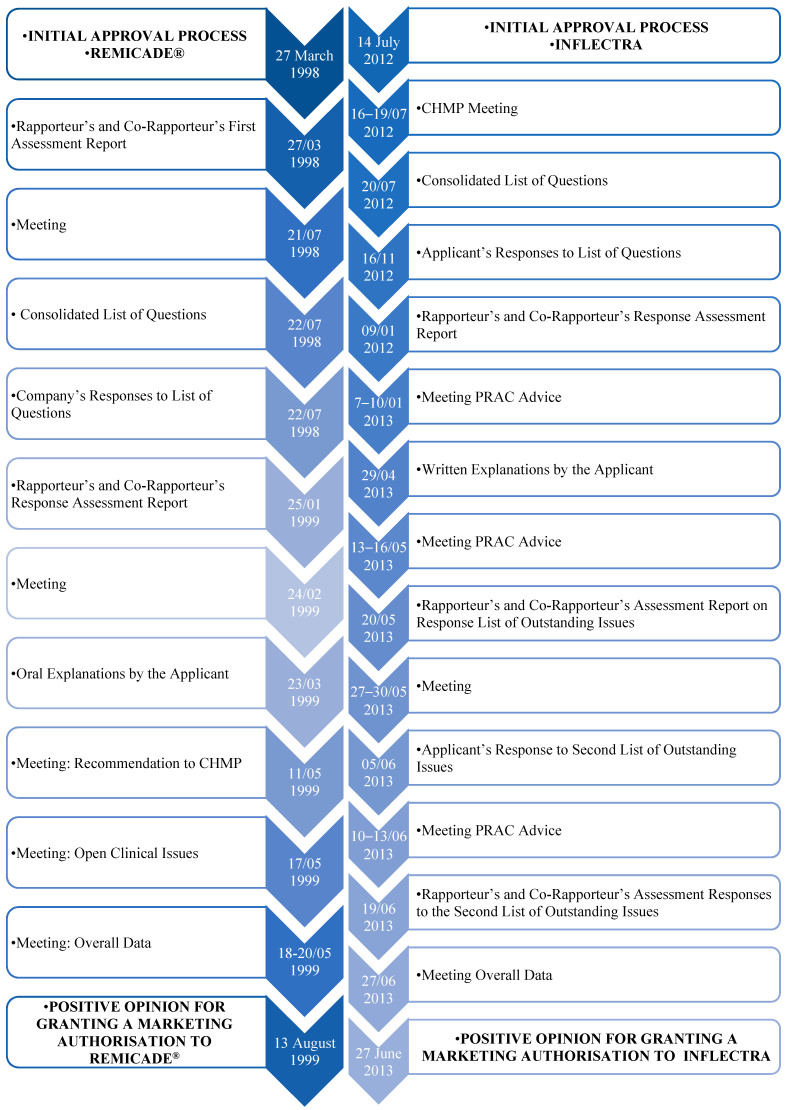
Approval process for Remicade^®^ and Inflectra, via EMA’s centralised pathway [77].

**Table 1 pharmaceutics-16-00321-t001:** Data requirements for the approval of a biosimilar and a reference medicinal product. (Adapted from [1]).

Reference Medicinal Product	Biosimilar Medicinal Product
Risk Management Plan	Risk Management Plan
Clinical studies: - Safety and efficacy - PK/PD - Immunogenicity	Comparative clinical studies: - Safety and efficacy - PK/PD - Immunogenicity Non-clinical comparative studies
Non-clinical studies	Comparative non-clinical studies
Pharmaceutical quality studies	Comparative quality studies: Pharmaceutical quality studies, namely, full data requirements regarding pharmaceutical quality, as well as additional quality studies comparing the structure and biological activity of the biosimilar medicine with the reference medicine.

**Table 2 pharmaceutics-16-00321-t002:** Critical quality attributes (CQA) and similarity demonstration products [12,40].

Comparability Critical Attributes	Similarity Demonstration
Protein structure and production quality	Extensive laboratory analysis for all molecular characteristics (multiple lots)
Pharmacokinetics, Pharmacodynamics, and Animal Toxicity	In vitro and in vivo assays (carried out on relevant animal species, if further confirmation scans are required for laboratory studies)
Pharmacokinetics, Pharmacodynamics, and Animal Toxicity	Phase I clinical studies
Clinical Efficacy and Safety	Phase III clinical studies
Safety in clinical practice	Risk management plan Phase IV studies Pharmacovigilance

**Table 3 pharmaceutics-16-00321-t003:** European guidelines for the development and evaluation of biosimilar medicines [52].

**Type of Guideline**	**Guideline Title**
General	Guideline on similar biological medicinal products.
	Guideline on similar biological medicinal products containing biotechnology-derived proteins as active substances: quality issues.
	Guideline on similar biological medicinal products containing biotechnology-derived proteins as active substances: clinical and non-clinical issues.
Specific	Guideline on biosimilar medicines containing recombinant granulocyte colony-stimulating factor.
	Guideline on clinical and non-clinical development of similar medicines containing low-molecular-weight heparins.
	Guideline on clinical and non-clinical development of similar medicines containing recombinant human insulin and insulin analogue.
	Guideline on clinical and non-clinical development of similar medicines containing beta interferon.
	Guideline on clinical and non-clinical development of similar medicines containing monoclonal antibodies: ethical and non-clinical issues.
	Guideline on clinical and non-clinical development of similar medicines containing recombinant erythropoietin.
	Guideline on clinical and non-clinical development of similar medicines containing the recombinant follicle-stimulating hormone.
	Guideline on clinical and non-clinical development of similar medicines containing somatropin.
Others	Comparability of biotechnology-derived medicinal products after a change in manufacturing process: clinical and non-clinical issues.
	ICH Q5E biological/biotechnology products subject to changes in their manufacturing processes: comparability of biological/biotechnology products.

**Table 4 pharmaceutics-16-00321-t004:** FDA’s development of a biosimilar medicine. Adapted from [13].

FDA advice on the scope and extent of testing during development	Highly Similar		The totality of the evidence
	Analytical Studies	Quality characteristics assessment, using state-of-the-art technologies and multiple different tests for the same characteristic, to determine whether the proposed biosimilar is highly similar to the reference product. Identification of differences in quality characteristics, if applicable, between the reference product and the proposed biosimilar (examples of quality characteristics may include structure and bioactivity). Evaluation of the potential impact of any observed differences.	
	Toxicity’s Assessment		
	Animal Studies	Support safety decision making before human exposure to the proposed biosimilar. Additional support to demonstrate similarity (but not always necessary).	
	No Clinically Meaningful Differences		
	Human PK and PD Studies	Comparison between the pharmacokinetics (exposure) and, if applicable, the pharmacodynamic (response) profiles of the proposed reference and biosimilar product to support a conclusion of similar efficacy and safety. Human PK and PD studies are generally considered to be the most sensitive data element to support a demonstration of no clinically meaningful differences.	
	Immunogenicity Assessment	Comparison between the incidence and severity of immune responses generated with the reference product and the proposed biosimilar. Immunogenicity assessment is generally included as part of all clinical studies.	
	Additional Clinical Studies	Additional clinical studies when residual uncertainties remain about demonstrating that there are no clinically meaningful differences after conducting the above-mentioned studies.	
	Experience with the Reference Medicine		
**Biosimilarity:** According to the general criteria defined by the FDA, the demonstration of biosimilarity is based on the totality of evidence provided when submitting a marketing application for FDA review. The presented data include an extensive analytical comparison in order to show that the proposed biosimilar medicine and the respective reference product are extremely similar in terms of structure and function. Moreover, animal, human pharmacological, immunological, and clinical data are enhanced as necessary to the analytical data, in a stepwise approach to provide the required information to demonstrate biosimilarity.			

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
