# Peer review of "A Regulatory Perspective on Biosimilar Medicines"

_pharmaceutics, 2024, doi:10.3390/pharmaceutics16030321_

Round 1

Reviewer 1 Report

Comments and Suggestions for Authors

The manuscript provides a good overview of the regulatory aspects of biosimilar products, which should be useful to the biotech and pharmaceutical industries. There is a lot of background information on biosimilars, which could be useful to someone who may be new to this area, but makes the manuscript very long (for example, Section 2.1 Generic and biosimilar drugs). Since the manuscript focuses on the regulatory aspects, it should be assumed that the readers have the background knowledge. This will shorten the manuscript. Specific comments are below:

1. Section 2. This section can be streamlined. I am not sure it is necessary to compare generic drugs and biosimilars.

2. Section 3. This section should be reorganized. The title may give the readers an impression that this section focuses on the manufacturing aspect. However, the subsections 3.1-3.5 are not related to manufacturing.

3. Section 3. lines 181-250 provide a lot of details on the manufacturing processes. This is not necessary if it is assumed that the readers have the background knowledge already.

4. Lines 253-256. This sentence is confusing and should be reworded.

5. Lines 278-281. This sentence is confusing and should be reworded.

6. Lines 285-290. This paragraph is an important discussion on immunogenicity. However, it is not clear whether what is discussed is the regulatory requirements. If yes, which agencies require them? 

7. Line 331. What does "exaggerated pharmacology" mean?

8. Line 366. What do you mean by "in vitro animal studies"? Are animal studies typically in vivo?

9. Section 5. I am not entirely sure about the purpose of this section. Is it intended to provide an overview of the biosimilar products that have been approved? Only Remicade and its biosimilar product are discussed. If this is a case study, it can be integrated into other sections, which should be more appropriate.

Comments on the Quality of English Language

May need some editorial help.

Author Response

REVIEWER #1

The manuscript provides a good overview of the regulatory aspects of biosimilar products, which should be useful to the biotech and pharmaceutical industries. There is a lot of background information on biosimilars, which could be useful to someone who may be new to this area, but makes the manuscript very long (for example, Section 2.1 Generic and biosimilar drugs). Since the manuscript focuses on the regulatory aspects, it should be assumed that the readers have the background knowledge. This will shorten the manuscript. Specific comments are below:

  1. Section 2. This section can be streamlined. I am not sure it is necessary to compare generic drugs and biosimilars.

R: Section 2 was streamlined accordingly. Section 2.1. Generic and biosimilar drugs was removed from the manuscript.

  1. Section 3. This section should be reorganized. The title may give the readers an impression that this section focuses on the manufacturing aspect. However, the subsections 3.1-3.5 are not related to manufacturing.

R: The title of Section 3 was reworded to 3. Development of Biosimilar Medicines in order to cover all described aspects in this section. Please refer to line 110.

  1. Section 3. lines 181-250 provide a lot of details on the manufacturing processes. This is not necessary if it is assumed that the readers have the background knowledge already.

R: Please note that this manuscript envisions to provide a practical perspective on the development of biosimilar medicines in which the manufacturing process assumes a critical importance towards the quality and reproducibility of the drug product. For that reason, particular details are provided in what concerns the major steps typically involved in the design and production of these products. This perspective also aligns with the comment addressed by Reviewer #4

  1. Lines 253-256. This sentence is confusing and should be reworded.

R: The sentence was rewritten accordingly and now reads: “Consequently, there exists significant potential for variations between an innovative biological medicine and a biosimilar medicine in terms of manufacturing processes”. Please refer to lines 186-188.

  1. Lines 278-281. This sentence is confusing and should be reworded.

R: The sentence was rewritten accordingly and now reads: “It is very unlikely for an adverse immune reaction to occur following the aforementioned modifications, as comparability studies indicate no significant increase in impurities or aggregates despite these alterations.” Please refer to lines 210-212.

  1. Lines 285-290. This paragraph is an important discussion on immunogenicity. However, it is not clear whether what is discussed is the regulatory requirements. If yes, which agencies require them? 

This discussion on immunogenicity was highlighted by EMA, as addressed in the mentioned reference. The cross-reference to EMA’s perspective on immunogenicity was introduced in the text. Please refer to lines 216-221.  

  1. Line 331. What does "exaggerated pharmacology" mean?

R: The sentence mentioning “exaggerated pharmacology” was rewritten and now reads: “The main mechanisms of adverse effects of biological drugs are related to exaggerated pharmacology. This phenomenon, known as on-target toxicity, entails the occurrence of adverse pharmacological effects specifically at the intended target site”. Please refer to lines 259-262. 

  1. Line 366. What do you mean by "in vitro animal studies"? Are animal studies typically in vivo?

R: The Authors thank to the Reviewer. The sentence was suitably corrected and now reads: “Generally, non-clinical data for a biosimilar is generated through an abbreviated testing programme or in vivo animal studies, as required by the EMA’s guidelines”. Please refer to lines 297-299.

  1. Section 5. I am not entirely sure about the purpose of this section. Is it intended to provide an overview of the biosimilar products that have been approved? Only Remicade and its biosimilar product are discussed. If this is a case study, it can be integrated into other sections, which should be more appropriate.

R: For a comprehensive list of the current commercialised biosimilars, the reader is guided to several reviews, particularly the ones carried out by the Purple Book published by EMA. Guided by the practical perspective on the development and regulatory path required for biosimilar evaluation/approval, the Authors present the example of Remicade and its biosimilars. More details were already introduced to make this analysis more clearly understood.

Comments on the Quality of English Language

May need some editorial help.

R: English was revised throughout the document in order to improve the overall quality.

Reviewer 2 Report

Comments and Suggestions for Authors

This manuscript provides an overview of biosimilar medicines with a focus on a regulatory perspective.  It discusses the guidance documents and regulatory frameworks.  This is an important topic.  However, the organization and clarity of the manuscript should be improved.  Specific comments:

1.      Highly similarity and no significant clinical differences from the reference product are the two key requirements for biosimilar products.  However, it seems the manuscript does not clearly discuss how to demonstrate these.  Please consider adding reviews on the current understanding of these matters.

2.      Page 4: the discussions on biosimilar products are redundant as they have been defined and discussed on page 3.

3.      Section 5: the approval processes for two biosimilar products are outlined in Figures 6 and 7.  The two figures may be combined for easy comparison.  Simple descriptions of the timelines may not be enough.  It would be suggested to add critical discussions on the two examples.  For example, based on the two successful examples, what can be known regarding development and approval, how biosimilar products can be successfully developed and approved, etc. 

4.      Research on biosimilar products is not really reviewed.  It seems the guidance documents and regulatory frameworks are the focuses.  However, the research is mentioned in Section 6 and Section 7. 

5.      Section 6 is confusing.  Organization and clarity should be improved.   

6.      Section 7 is relatively long and less concise.  It does not seem to be a conclusion. 

Author Response

REVIEWER #2

This manuscript provides an overview of biosimilar medicines with a focus on a regulatory perspective.  It discusses the guidance documents and regulatory frameworks.  This is an important topic.  However, the organization and clarity of the manuscript should be improved.  Specific comments:

  1. Highly similarity and no significant clinical differences from the reference product are the two key requirements for biosimilar products.  However, it seems the manuscript does not clearly discuss how to demonstrate these.  Please consider adding reviews on the current understanding of these matters.

R: The analysis on the key requirements for biosimilar products were considered in the manuscript. Moreover, reviews in this scope were added to the current understanding of these matters. Please refer to lines 94-109.

  1. Page 4: the discussions on biosimilar products are redundant as they have been defined and discussed on page 3.

R: In order to avoid redundant information, and in line with Reviewer #1’s comment, Section 2 was streamlined accordingly. Section 2.1. Generic and biosimilar drugs was removed from the manuscript.

  1. Section 5: the approval processes for two biosimilar products are outlined in Figures 6 and 7.  The two figures may be combined for easy comparison.  Simple descriptions of the timelines may not be enough.  It would be suggested to add critical discussions on the two examples.  For example, based on the two successful examples, what can be known regarding development and approval, how biosimilar products can be successfully developed and approved, etc. 

R: Figures 6 and 7 were combined as suggested. Please refer to lines 682-709. Moreover, captions were considered in this section in order to improve the organisation of the section and for more clear reading. Please refer to lines 577, 625, 651 and 674.

  1. Research on biosimilar products is not really reviewed.  It seems the guidance documents and regulatory frameworks are the focuses.  However, the research is mentioned in Section 6 and Section 7.

R: Section 6 reflects on the challenges, benefits, risks and relevant particularities regarding biosimilar medicines while section 7 intends to summarise the concepts studied throughout this review, as well as a global perspective on the importance and impact of this type of medicines. Research on biosimilar medicines is not considered an aim of this work, however, several articles/reviews have been referred throughout the manuscript to suggest the readers a further analysis.

  1. Section 6 is confusing.  Organization and clarity should be improved.   

R:  Section 6 was reorganised in order to provide a more clear perspective for the readers. Please refer to lines 710-803.

  1. Section 7 is relatively long and less concise.  It does not seem to be a conclusion. 

R: Section 7 was shortened. Please refer to lines 804-856.

Reviewer 3 Report

Comments and Suggestions for Authors

The review discusses biosimilar medications from the definition, production regulatory issues, and market perspective, which is potentially interesting for a broad range of readers as a new, fast-developing field in pharmacy.

Lines 47-57 The review's aims are much wider than the title suggests; change the title to meet the whole review topic.

Lines 181-185 The claim that biosimilar development costs a hundred times more than generic drugs is not supported by data in the table or reference; please change the claim or support it with appropriate reference.

Lines 422-426: Please state the statement in this sentence, as the joint statement is available online.

Line 492 PGA abbreviation is not defined before.

Lines 667-669 Please support the claim of a growing market by reference. Intuition says it is true, but a reference is needed to support the claim or rewrite the claim.

Line 718 Change the capitalization of the drug name.

General comment: The review is well-written and structured; while reading, I missed more details on protein structure similarity and its importance in regulating biosimilar medications. Please improve.

Comments on the Quality of English Language

Please check the usage of articles, commas, and prepositions. Avoid the wordy expressions, and simplify your writing as much as possible. 

Author Response

REVIEWER #3

The review discusses biosimilar medications from the definition, production regulatory issues, and market perspective, which is potentially interesting for a broad range of readers as a new, fast-developing field in pharmacy.

 Lines 47-57 The review's aims are much wider than the title suggests; change the title to meet the whole review topic.

R: The review’s aim was rewritten in order to make clear its purpose and align with the defined title “A Regulatory Perspective on Biosimilar Medicines”. Please refer to lines 45-56.

Lines 181-185 The claim that biosimilar development costs a hundred times more than generic drugs is not supported by data in the table or reference; please change the claim or support it with appropriate reference.

R: The claim was duly supported with the appropriate reference. Please refer to lines 111-114.

Lines 422-426: Please state the statement in this sentence, as the joint statement is available online.

R: The statement was stated in this sentence and duly supported with the appropriate reference. Please refer to lines 362-370.

Line 492 PGA abbreviation is not defined before.

R: The Authors thank to the Reviewer. The abbreviation was suitably corrected to “FDA” and now reads: “FDA has only recently started formulating distinct guidelines for the approval of bio-similar drugs”. Please refer to lines 434-435.

Lines 667-669 Please support the claim of a growing market by reference. Intuition says it is true, but a reference is needed to support the claim or rewrite the claim.

R: The claim was rewritten and duly supported with the appropriate reference. Please refer to lines 572-575.

Line 718 Change the capitalization of the drug name.

R: The drug name was changed accordingly and now reads: “Remicade®”. Please refer to line 626.

 General comment: The review is well-written and structured; while reading, I missed more details on protein structure similarity and its importance in regulating biosimilar medications. Please improve.

R: Details on protein structure similarity and its importance in regulating biosimilar medications were added in the manuscript. Please refer to lines 328-336.

Comments on the Quality of English Language

Please check the usage of articles, commas, and prepositions. Avoid the wordy expressions, and simplify your writing as much as possible. 

R: English was revised throughout the document in order to improve the overall quality.

Reviewer 4 Report

Comments and Suggestions for Authors

This manuscript is aimed to reflect on and understand the procedures regarding the  regulatory framework for biosimilar medicines. Manuscript shows a global perspective on the importance of this type of medicine, the impact they have on the health and well-being of individuals and society, ending with possible associated future perspectives. Explanation of the biosimilar manufacturing and production process is given. Manuscript can be accepted after minor revision. My comments are below:

- Definition of biosimilar medicinal products should be more clear.

- More examples of biosimilars should be given.

Author Response

REVIEWER #4

This manuscript is aimed to reflect on and understand the procedures regarding the  regulatory framework for biosimilar medicines. Manuscript shows a global perspective on the importance of this type of medicine, the impact they have on the health and well-being of individuals and society, ending with possible associated future perspectives. Explanation of the biosimilar manufacturing and production process is given. Manuscript can be accepted after minor revision. My comments are below:

- Definition of biosimilar medicinal products should be more clear.

R: The definition of biosimilar medicines is referred in lines 58-63. Nevertheless, it was made clearer by addressing the key requirements such as highly similarity and the absence of significant clinical differences from the reference product. Please refer to lines 94-109.  

- More examples of biosimilars should be given.

R: Readers are directed to references that have already exhaustively reported biosimilars already marketed. Please note that reinforced analysis is now provided for the case study presented.

Round 2

Reviewer 2 Report

Comments and Suggestions for Authors

n/a